# Workplace Physical Activity Barriers and Facilitators: A Qualitative Study Based on Employees Physical Activity Levels

**DOI:** 10.3390/ijerph19159442

**Published:** 2022-08-01

**Authors:** Ayazullah Safi, Matthew Cole, Adam L. Kelly, Mohammed Gulrez Zariwala, Natalie C. Walker

**Affiliations:** 1Centre for Nutraceuticals, School of Life Sciences, University of Westminster, London W1W 6UW, UK; m.zariwala@westminster.ac.uk; 2Centre for Life and Sport Science (C-LaSS), Birmingham City University, Birmingham B15 3TN, UK; matthew.cole@bcu.ac.uk (M.C.); adam.kelly@bcu.ac.uk (A.L.K.); 3School of Life Sciences, Coventry University, Coventry CV1 2DS, UK; ad9487@coventry.ac.uk

**Keywords:** qualitative research, physical activity, workplace, barriers, facilitators, employees health and wellbeing

## Abstract

Introduction: Lack of time, management support, insufficient facilitates, workload balance, and culture are often reported as common barriers to physical activity (PA) participation in the workplace. In comparison, identifying facilitators of PA in the workplace are scarce. A ‘one-size-fits-all’ approach to overcoming the barriers may also be unsuccessful within university settings where multidisciplinary workforce exists due to the heterogeneity nature of job roles. Thus, the aim of this study was to understand the perceived barriers and facilitators of PA of university employees who were classified as active or inactive based on their job roles. Methods: Forty-one employees (female = 17; male = 24) participated in focus groups to discuss their perceived barriers and facilitators to PA in the workplace. Participants were categorised based on their PA levels as active and inactive prior analysing the semi-structured focus groups data via using thematic analysis. Results and Discussion: The results showed that a lack of time was reported by 80% of the participants as a barrier to PA, including 63% inactive and 17% of the active participants. This included 27% administrators’ staff, 23% academics, 19% senior management, and 11% professional service staff. Over 75% participants reported a lack of management support as one of the perceived barriers to their PA engagement in the workplace. Approximately 58% also reported workplace culture as a barrier to PA participation. Open access to a gym on campus was perceived to be the main facilitator to engaging in PA in the future. Similarly, increased management support for engaging in PA and having flexibility during working days were perceived as facilitators for PA engagement and a way to reduced sedentary behaviour in the workplace. Conclusions: These findings contribute to the limited literature in terms of evaluating obstacles and facilitators of university employees to encourage engagement with PA in the workplace. These findings can be applied to form PA, health, and wellbeing-related interventions specifically targeting these identified barriers that are experienced in the workplace and thereby potentially reducing absenteeism and increasing productivity.

## 1. Introduction

Participating in regular physical activity (PA) has been reported to aid in the prevention of many chronic diseases, including cardiovascular disease, type II diabetes, colon cancer, hypertension, and musculoskeletal disorders [1,2,3,4]. PA participation also enhances feelings of wellbeing [5] and self-esteem [6] and decreases anxiety and depression [7,8]. Despite the wide-ranging benefits, according to the World Health Organisation [9], over 60% of the population do not meet the recommended PA guidelines. Thus, they may not receive the health benefits associated with PA participation. Most people failing to meet the recommended PA guideline is partially due to an increased sedentary behaviour during concerning a comprehensive approach for promoting PA and health especially in the workplaces [10]. Workplace policies, culture, and staff flexibility are reported to have a major effect on employees PA levels and health [11]. Research indicates that employees spending approximately 60–70% of their waking hours in the workplace, with over 75% of this time spent being sedentary [12,13,14]. Physical inactivity can improve if policymakers, public health researchers, and leisure recreation sectors embrace challenges and provide PA and health related interventions in the workplace [15].

Research also suggested that workplaces supporting PA, health and wellbeing programmes observed benefit in various ways, including improved productivity, higher staff retention, greater loyalty, and reduced sickness absences [16]. However, there is a mixture of findings without clear guidelines to understand which interventions were most suitable to be effective in this environment. For instance, most interventions failed to identify baseline PA levels, sedentary behaviour, and health and did not consider the fundamental purposes of employees engagement and disengagement in the workplace PA and health interventions [17]. Despite inconclusive findings of interventions, previous research has suggested that the common reasons for employees not engaging in PA are lack of time because of workload and no access to resources [18,19]. In addition, lack of knowledge, workplace culture, management support, and low self-motivation are also reported as barriers to PA engagement across job roles from different disciplines [20,21,22,23]. Most of the previous literature has mainly focused on identifying barriers to PA but is limited in their proposals of exploring what would actually facilitate engagement in PA in the workplace. Thus, workplace barriers may require a holistic approach as staff face various pressures and responsibilities related to their job roles that could limit their PA participation. A ‘one-size-fits-all’ approach may remain unsuccessful within a multidisciplinary workforce such as university setting. Hence, there is a need for exploration to identify the key barriers to PA in university settings and to specifically ask employees how these can be overcome and their preferred facilitators that might support tailored PA and health-related interventions to promote PA engagement in the workplace [24,25,26].

Promoting PA levels in the workplace is pertinent because employees spend approximately 60–70% of their waking hours in the workplace, with over 75% of this time spent being sedentary [12,13,14]. In addition, it is also worth noting that most of the previous literature has applied quantitative methods for investigating employees’ barriers to PA, hence missing detailed insight because quantitative and objective methods provide information about how much and where, whilst qualitative methods provide an insight into why and how [27,28]. Previous research supported the use of a qualitative approach, as it helps provide an insight into the lived experiences and challenges people are facing [29,30]. Of the previous research that has used a qualitative approach their sampling frames were non-focused, and participants discussed PA participation in future interventions, which had not been offered [30]. Thus, future research is needed to explore barriers to PA and health across different job roles [31]. Qualitatively exploring barriers and facilitators can provide detailed insight into participant’s feelings, beliefs and attitudes in a way that may not be feasible in other methods [27,28]. Thus, the aims of this study were to understand the perceived barriers and facilitators of PA, health, and wellbeing of university employees through focus groups. An additional aim was to explore if different barriers and facilitators were identified based on job roles and genders and what types of PA, health, and wellbeing initiatives individuals might commit to participate.

## 2. Methods

### 2.1. Participant Recruitment

Following an institutional ethical approval, participants were recruited via an opportunistic sample using those who had participated in earlier studies of a broader piece of research from a UK higher education institution (university workplace) [32,33]. From the original pool of 259 participants who signed up to be contacted, 221 employees (85%) replied expressing their interest in the present study. Despite sending four reminders, one email per week, 180 employees did not return their consent forms. Table 1 provides the breakdown of total participants who attended a focus group interviews. The inclusion criteria were that all participants had to be adults (>18 years old), currently employed by the participant university, and had participated in earlier studies [32,33].

Activity categorisation was based on previous studies where participants completed the International Physical Activity Long Form Questionnaire (IPAQ-LF), wore an ActiGraph for a whole week and were deemed active if they had met the WHO PA guidelines of achieving at least 150 min of MVPA [9]. Inactive participants were those who did not meet the WHO PA guideline [32,33].

Categorising participants into groups according to their PA levels was vital. It provided an opportunity to compare active and inactive employees to explore differences and reduced potential bias in responses provided. Analysing the perceptions from these two separate groups may provide a comprehensive insight into the actual barriers and facilitators that this population is facing that is not influenced by the experience of the other group. Participants were not informed specifically about their PA coding category for the purpose of this study.

### 2.2. Research Procedures

A qualitative method was applied, as this was deemed a suitable approach for understanding employees’ lived experiences and identifying perceived barriers and potential facilitators to PA. The adoption of focus groups was done specifically to aid comparing active and inactive individuals’ barriers to PA and support the sampling size. Utilising a focus group approach, as opposed to individual interviews, may have influenced a sense of camaraderie in the group. Previous research proposed that a focus group is an enjoyable experience and provide an opportunity for individuals to hear others’ perceptions regarding the same issue [34,35]. Several sensitive and personal disclosures were likely to arise in a focus group setting, with some sensitive themes compared to individual interviews [36]. Thus, this study used focus groups to understand the specifics of collective barriers and facilitators to PA specific to active and inactive participants.

Previous research also suggests that focus groups are appropriate in settings where power differences exist between participants and decision makers, mainly when daily use of language and culture of the under-researched population is of interest and requires exploring organised issues [37]. This applies to the university settings with a variety of job roles and levels featuring within this sample. Therefore, eight semi-structured focus groups with an average duration of the interview lasting 35 min per focus group took place in two different campuses and within four faculties of one university. Participants held various job roles and levels such as academics, senior management, professional services, and administration staff. The focus groups were conducted to gain an insight into respondents’ attitudes, feelings, beliefs, and experiences related to barriers, facilitators and desire for PA and health initiatives.

A minimum of five employees participated in each focus group in accordance with previous literature [38]. The use of this number of participants means that rapport can be built, which affords more opportunity to share ideas [38]. Therefore, this was ensured that all focus group participants aligned to the same activity level (i.e., active, or inactive). The same set of questions were asked of all focus groups except for any probing questions based on the information from participants. The questions guide was developed according to the focus group methodology [39]. For the research process see Figure 1. 

### 2.3. Pilot Study 

Previous research suggested the importance of pilot studies in general but piloting in qualitative research is limited [40,41,42]. The pilot study provides an opportunity to modify, review approaches and improve experience with adopted techniques [43]. Therefore, prior to the actual focus groups, two pilot focus groups were conducted in this study. A total of 13 critical peers (male n = 7; females n = 6) participated in the pilot study. Participants were divided into two groups; active participants (n = 7: 4 males; 3 females) and inactive participants (n = 6: 4 males; 2 females). The pilot focus groups were conducted with staff who were not the actual study participants.

### 2.4. Thematic Analysis

The semi-structured focus group data were analysed using thematic analysis (TA). The focus groups were transcribed verbatim, checked, and coded according to the framework by [44]. TA aims to identify themes that are important for exploring specific issues [45]. TA is a frequently used technique in qualitative research as it provides a systematic and transparent structure for the data analysis [46]. Previous research has supported the use of TA as the appropriate approach to explore issues [46,47]. Previous research suggested that TA provides implicit and explicit analysis, identifies ideas and themes that emerged from the data [44]. Thus, TA was considered appropriate for this study because it provides clear perspectives of the issues and allows to combine themes associated with each other [46]. With regard to the trustworthiness of results, this study used the most common trustworthiness model consisting of five conditions: credibility, dependability, conformability, transferability, and authenticity, combined to construct trustworthiness [48].

During the focus groups, the questions were introduced one by one to receive an adequate response from participants. The lead researcher moderated the conversation and used prompts for clarity or requests for further information when required as per previous research [49]. Taking this approach-built group reciprocity created an intimacy; therefore, sensitive research topics (i.e., lack of PA in the workplace) could be discussed in-depth, and participants were encouraged to share and provide supporting or contrasting statements until no new themes emerged. The researcher influence was minimised by opting to remain silent during the discussions, infrequently prompting the group when discussions reached stagnation and then followed with further questions as per the previous research [50]. Thus, the saturation was reached during focus groups, and the data in this study were member checked from the raw data through to complete analysis for data saturation.

### 2.5. Results and Discussion

This section reports the results and discussion explicitly relating to the shared barriers and facilitators for active and inactive participants (Table 2), followed by those only evidence for the active participants (Table 3), and then data specific to only the inactive participants (Table 4).

### 2.6. General Barriers to PA

Regarding the general barriers, 80% of the participants, including 63% inactive and 17% active groups, reported lack of time as a barrier to PA engagement. This includes around 27% of administration staff, 23% academic staff, 19% senior management, and 11% professional services staff. It is also worth noting that 67% of female employees who participated in this study reported lack of time being a barrier to their PA participation. Lack of time was the commonly reported barrier to PA participation between employees across job roles too, which could be due to their perceived or actual workload. The present findings support previous research regarding lack of time being a commonly reported barrier to PA engagement [17,51]. Moreover, 29% of participants suggested that the weather, and more specifically the winter’s darkness, prevents them from engaging in PA. Some employees noted that they are likely to engage in PA (e.g., walking or jogging), but it is often difficult due to the UK’s poor weather conditions. The present findings align with previous research concluding that the extreme weather, including cold temperature and the winter’s darkness, negatively affects PA participation [52]. This might be a more significant concern for females, whereby 57% of females in the present study reported weather as a barrier to PA.

Forty one percent of the participants suggested family as a barrier to their PA engagement. This included 29% of inactive and 12% active participants. In total, 15% of academics, 11% administrator staff, 10% professional services staff, and 5% senior management outlined family as barrier to PA engagement. This was particularly salient for female participants, with 70% reporting this as a significant barrier. The present findings support previous research suggesting that family burden, particularly childcare responsibilities, preventing them from engaging in PA [53,54]. Universities are diverse organisations with a range of job responsibilities; thus, participants perceived family commitments as restricting them from being active because they spent two to three days away from home. When they return, they try to spend as much time with their family, and most of that time is spent being sedentary. Therefore, if workplaces provide opportunities to employees to engage in regular PA whilst at work, this could be regarded as a reward and would allow employees to spend more time with family, and this could lead to a happy and productive workforce. An example of how family can impact on PA is illustrated below: 


*“I live an hour and a half away from work; I stay 2–3 nights away from home every week because I am away most of the week, so when I go home, I try to spend as much time with family/children as possible and most of that time is spent sitting because I am tired, and this affects my family/children too”*
(Administrator staff).

This barrier may negatively impact employees PA engagement, health, or wellbeing and influence family and children’s lifestyle. For instance, family support is essential for an active and healthy lifestyle, and this is evident in the previous research suggesting that family/parental support found to be helping family/children meeting the PA and health behaviour guidelines, such as reduced screen time and vegetable consumption [55,56]. Previous research applied the Eccles expectancy theory to examine the influence of family/parental support regarding children PA [57]. The results revealed that children of active families/parents were also active. To facilitate family-related barriers, the influence of family support must be considered, not merely for staff, but also for their family/children, as it plays an important role in developing an active and healthy lifestyle [58]. Finally, a lack of motivation was also noted as a barrier to approximately 51% of participants. This included 43% inactive and 8% active participants. Furthermore, 19% professional service, 14% administrator, 11% senior management, and 7% academic staff reported this theme as a barrier to PA participation. This finding aligns with previous research suggesting that lack of motivation was one of the most commonly perceived barriers to PA participation [59,60]. Employees reporting lack of motivation as a barrier could be due to their lack of intrinsic motivation. However, lack of management support and working environment can also be detrimental and lead to stress, burnout, and anxiety amongst employees’ [61].

## 3. Workplace Barriers to PA

Regarding the workplace barriers to PA, approximately 58% of the participants reported workplace culture as a barrier to PA participation. This included 26% of administrators, 18% professional services, 9% academics and 5% senior management staff. Thirty-seven percent of the inactive participants and 21% of the active participants reported this theme. Moreover, over 75% of the participants reported lack of management support as one of the perceived barriers to their PA engagement in the workplace. This was particularly salient in the inactive participants (67%) and female participants (63%). Additionally, 51% suggested that short lunch breaks and adequate flexibility were barriers to their PA Participation. This appeared to be particularly salient for administrator’ staff (25%) in comparison to 19% professional services, 5% senior management, and 2% academics. The present findings are similar to previous research suggesting that physical inactivity culture and lack of flexibility are barriers to PA engagement in a typical workplace [17]. Universities are diverse environments regarding job roles, power differences, and gender. Thus, the existence of different workplace barriers within this workplace may be apparent. The lack of incentives, inflexibility, and limited management support are also identified in the literature as one of the main barriers to PA engagement within the university setting [62]. Thus, present findings regarding working flexibility and management support previous research. In this study, academic employees are perceived to have more flexibility compared to other staff (i.e., administration staff), and this could be a reason why many participants in the current study from other job roles suggested this as a perceived barrier, as illustrated by examples below: 


*“If flexibility was there, for example, we can stay until late or leave early to attend the exercise classes would be really appreciated”*
(Administrator staff).


*“Depends on your role whether you got that flexibility. The staff need to have flexibility. I think staff need a flexible approach”*
(Professional staff).


*“I would be able to take a long enough break to have a swimming session”*
(Senior lecturer).

Of the 85% participants, 63% inactive and 22% active reported lack of resources as a barrier to PA participation in the workplace. Employees stated that changing facilities, including showers, were inadequate. Participants suggested that if staff had an active commute to work (e.g., walk, jog, cycle) then they would need appropriate facilities to then commence the working day, and without the facilities, it was a demotivator to participation. The number of staff and students at this university means there is a real need for appropriate facilities if PA is to be promoted. Overall, the lack of facilities appeared to be one of the common barriers across all participants. This may be one of the critical barriers preventing employees from choosing a healthy and active lifestyle in the workplace, as illustrated below. 


*“I think the university of any size have decent sports and shower facilities. Our university with 25000+ students and around 2000 staff, and we have one shower at the City Centre campus. Surely something needs to be done about this, because look if there is no sports and exercise facilities and no showers, I mean yeah, it’s a real struggle. Yes, I exercise outside of the Uni because there are no facilities here or when I cycle or walk to work, I am soaked (sweating), and I need to have a shower. There is one shower, and that’s on the 3rd floor, and that really puts me off”*
(Professional service).

### Facilitators of PA

Whilst facilities have been identified as a barrier to PA engagement, the participants reported some specific facilitators linked to resources. This included having open access to a gym on campus, which may overcome employees’ barriers to PA. Staff would be more likely to be motivated, and this would lead them to engage in PA/exercise if access to resources were provided such as open access to gym onsite. This was reported by 33% of the administrators, 29% of professional services, 21% of academics, and 7% senior management. Additionally, over 65% of the total participants (53% inactive and 12% active) perceived that activities including dancing, yoga, walking, running, and competitive sports would facilitate their engagement in regular PA in the workplace. Some quotations from employees as examples are illustrated below: 


*“We were under the impression that there would be a gym, but it never happened. Someone needs to be an entrepreneur and open a gym on campus, and I think it will do well. I joined a gym near my house, and it is another university’s gym, as we don’t have anything like it here, so university here needs to open a gym”*
(Associate professor).


*“The University invested a lot of money on Doug Ellis, but now everybody is relocating, and it’s a shame that nothing like that is brought down here (Edgbaston campus). Staff and students would really appreciate a gym with easy access near here (City Centre campus). I think a gym would make the university more attractive as well”*
(Administrator staff).


*“I am really confused like we have a brand-new building, but there isn’t a gym for staff or students, I mean, that is ridiculous. Other small colleges and universities have a gym, and why can’t we have it here on Campus”*
(Assistant lecturer).

Previous research has discussed the efficacy of providing walking clubs, gym access, and activities ranging from dancing, yoga, counselling, and nutrition. The findings suggest that access to a gym and walking clubs effectively improved PA engagement in the workplace [51,63]. In addition, employees reported an overall satisfaction and suggested that their workplace is taking care of their health and wellbeing [51]. Employees reporting satisfaction could lead to retention, loyalty, reduced absenteeism, and improved productivity, which, in turn, can benefit the workplace.

A shared recommendation from active and inactive groups suggested that participant universities need to consider a flexible policy and allow employees longer breaks or work from home to promote PA engagement as illustrated below: 


*“I think you need more than 30 min break. We tend to be in academic services our hours are set like 9-5 and have 30 min lunch, and you can’t do PA in 30 min of break, this workplace is not flexible, and even if we had a flexible policy like working from home, sometimes this will help most staff go for a walk or do some sort of activity at home”*
(Professional services).


*“If this university was flexible and allowing us to be away from the desk for longer or work from home at least two days a week, it would help many staff do more PA”*
(Administrator staff).

The need for flexibility was proposed by 46% of participants. This includes 33% of the inactive groups and 13% of active groups, with 37% female participants suggesting this need. In this study, not having sufficient flexibility was more of a barrier to females than males, and this is not only related to the workplace but also to the families. For instance, this was particularly relevant for female participants, with 70% reporting family as barriers to PA participation because of the perceived family commitments, including childcare responsibilities restricting them from being active. Previous research evaluated the impact of flexible working policies on employees, such as working from home and the subsequent impact on PA levels and sedentary behaviour [64]. The results suggested that flexible working policies did not change PA levels (Z = −0.29, *p* > 0.05), and surprisingly, sitting time was recorded higher on days when staff worked from home (Z = −2.02, *p* > 0.05) compared to in office (Z = −4.16, *p* > 0.001). Moreover, the COVID-19 pandemic led around 88% of organisations globally implementing home-based working leading to an unprecedented transition into a sedentary working lifestyle [65]. Therefore, PA engagement may not increase with flexible policies such as working from home, but if workplaces consider alternatives such as providing access to the gym or other PA and health-related initiatives and resources to employees at work with flexibility to engage in PA during working hours could have a positive impact on staff PA engagement. The following section is focused on the results and discussion exclusive to each group such as active and inactive, commencing with Table 3, and then discussing the results of the active group followed by the inactive group in Table 4, respectively.

## 4. Discussion of the Barriers to PA Specific to the Active Group

In total, 35% of participants in the active group, including 15% academics, 13% professional services, and 7% administrator’ staff, suggested that road safety was a barrier to their participation in PA. The current findings support previous research where road safety has been highlighted as one of the barriers to PA as individuals feared for their health and safety due to lack of cycling routes in the United Kingdom [66,67].


*“I started cycling but there isn’t a safe route for cycles, and I wouldn’t start again until there is safe roads because I know how bad people drive”*
(Assistant lecturer).

### 4.1. Facilitators of PA Specific to the Active Group

The aspects that would further facilitate PA in the workplace active group individuals included personal factors such as perceived enjoyment (45%) and adopting a positive attitude towards PA (25%). Some participants suggested that they are using weekends to exercise to compensate for their workplace weekly sedentary behaviour. This shows that employees can be self-motivated and have a positive attitude to PA when they have time and autonomy. As highlighted by self-determination theory, the three main requirements for human beings are autonomy, competence, and relatedness. One of the critical necessities for a human is autonomy, such as being self-directed or being in control [68]. When individuals have freedom, it can lead to motivation and satisfaction, and in turn, this results in better wellbeing [69,70]. Interventions focused on promoting PA, health, and wellbeing in the workplace could improve PA engagement if autonomy is considered. Moreover, participants in this study reported that they enjoy partaking in PA and suppose they miss the opportunity to engage in exercise. In that case, they even miss it mentally because they are aware that being sedentary is harmful to their health and wellbeing, as illustrated below: 


*“I run, and I enjoy it and I been doing it for two and half years, and I am happy to continue doing it, but if I miss it, I miss it, and I miss it mentally”*
(Senior lecture).

When an individual identifies being physically active as a favourable substitute for being sedentary, it is regarded as the counter conditioning process, which could lead to self-re-evaluation in self-re-appraisal, such as that their sedentary behaviour in the workplace is making them feel disappointed (Adams and White, 2004). This behaviour can lead to the maintenance stage of the Transtheoretical Model (TTM), which may yield a healthy and active lifestyle [71]. Moreover, the Social–Ecological Model (SEM) suggests that people’s behaviour/attitude is not only influenced by the intrapersonal characteristics but also by social factors including the interpersonal community, an organisation such as the policies, rules and overall environment influence behaviours, attitudes and knowledge that may impact their health [72]. The subsequent section is focused on the results that are unique to only inactive employees.

### 4.2. Discussion of the Barriers to PA Specific to the Inactive Group

This group also suggested a range of barriers preventing them from engaging in PA. For instance, 90% of inactive participants including 37% administrator staff, 26% academics, 23% professional services and 4% senior management reported that they have an inactive commute to work and perceived this as a barrier to their PA participation. Overall, 73% of female employees reported an inactive commute to the workplace. Participants choosing inactive commute could be due to the distance, lack of education or lack of time, as reported being the key perceived barriers to PA engagement in this study. Educating or being aware of the health and wellbeing benefits related to active commute may lead people to choose an active or a combination of active commute to the workplace. Previous research compared active and inactive commuters and concluded that people who were aware of the associated benefits of active commutes reported positive effects on mood states, energy level, better health, and enhanced productivity compared to inactive commuters [73]. For instance, in this study, most inactive employees were not aware of the recommended PA guidelines. Thus, this may have been one of the contributing factors to inactive commute such as driving, which affects PA levels, increases sedentary behaviour, and may negatively impact their health and wellbeing. For instance, the TTM views behaviour change as a development course rather than a single event, and it determines that people move through five stages as their behaviour changes from unhealthy to healthy [74]. Hence, behaviour change takes place through stages, and participants in this group noted they mainly take inactive commutes such as driving and parking near the office to avoid walking. This is known as the precontemplation stage of the TTM, in which individuals may not be conscious of the impact of their behaviour on their health and well-being, and if they become aware their behaviour, they are more likely to move to the contemplation stage [75].


*“I commute to work in the car, and then I try to find a car park space near my office, so I don’t have to walk much”*
(senior management).

Moreover, environment was reported as a barrier by 42% of the inactive participants, including 27% of females and 21% in professional services, 13% administrators, 5% academics, and 3% senior management staff. Workplace or community sectors providing opportunities for individuals to engage in PA are regarded as supportive environments, playing a positive role in encouraging active behaviour. However, an unsupportive environment, including fear for safety, is regarded as intermediated, leads to inactive behaviour, and is considered a barrier to PA participation. Recent research suggested that individuals working or residing in an environment in which PA is regarded as necessary are more likely to perceive fewer barriers to PA, hence engaging in regular PA compared to those in an environment where PA is not a social norm [76].


*“There are canals to run, and I been there with my students, but I wouldn’t recommend it for a male or female. It’s pretty intermediated, and there are security issues with the canals; it’s not a fantastic place to go there, especially alone”*
(Senior lecturer).

Tiredness, busy lifestyle, costs, lack of like-minded people, and workplace pressure were also reported as perceived barriers to PA participation amongst this population. Tiredness was reported as a perceived barrier to PA by approximately 71%, including 37% of male and 34% of female employees. Being too tired to participate in PA was similarly observed within this population regardless of their gender. This could be due to workplace pressure, busy lifestyle, and participants not being surrounded by like-minded individuals such as colleagues and family/friends to encourage PA participation either within or outside of the workplace. The current findings align with previous research suggesting that sensitivity to work colleagues’ behaviour may well reflect a broader need to be perceived as a barrier to PA participation [76,77].


*“I went for a walk with colleagues once and yeah there weren’t many likeminded people, and I didn’t go again”*
(lecturer).

The cost of PA was also reported as a particular barrier for the inactive employees (62%), especially by females (69%), administrators (31%), and professional services (54%) compared to academics (31%) and senior management (15%). This could be due to their salary differences that academics and senior management typically tend to be on a larger scale than other job roles. Previous research has suggested an association between socioeconomic status (SES) and PA participation, as people from higher SES tend to be more physically active compared to the those of lower SES [78,79,80]. This could be applicable to employees in this study, as they were from a range of job roles, and despite showing interest, some participants were already engaging in PA but gave up because of the cost. Thus, if the workplace provides facilities including open access to gym and other health and wellbeing related activities in the workplace, this could help employees from across job roles to be physically active.


*“Yeah, the cost is a big one for me. I used to swim once a week, but I can’t anymore, and there aren’t like-minded people to go with, and my family isn’t here. I am tired after work as well, so yeah, I mean that affect my motivation to go all the way to another university and use their swimming pool”*
(Professional service).

Although the participant university has an affiliated membership for staff to use another university’s exercise facilities at a discounted rate, most of the employees appeared not to have engaged with this opportunity because of the cost.


*“Well, the cost is a big factor when I go to the swimming, although I have the associated thing in Aston pool it does bring the cost down but still it’s not cheap, and I don’t think I can subscribe to a long term because I don’t know if I can commit to it cost-wise as well as because of all the other reasons we just mentioned”*
(Lecture).

The present findings align with previous research suggesting that people from lower SES reported lack of PA engagement [81]. As detailed in the SEM, the organisational rules, regulation, and strategies can either promote or endanger employees’ health [72].

### 4.3. Discussion of the Facilitators to PA Specific to the Inactive Group

Approximately 76% of participants in this group suggested that having autonomy would facilitate their PA engagement, and 67% suggested that having a protected time for exercising in the workplace would help them engage in PA behaviours. This was particularly salient for the administrative and professional services (33% and 21%, respectively). Moreover, participants suggested that protected time would help them move away from their desks and be more active. This university had entry passes for the ‘Botanical Garden’, which might be used for 15–20 min of a day, but these could only be utilised if the staff are encouraged to move away from their desks.


*“It’s too tempted to sit at a desk and have lunch. It’s simple rather than get out at lunchtime and walk is too difficult. If there were a protective time to exercise, that would have been helpful; even having passed to the Botanical Garden for 15–20 min walking in a garden would actually help a lot. However, that time has to be protective so staff can freely move away from their desk without worrying”*
(Lecturer).


*“There should be some protective time during the day if staff wanted to do some exercise or something else that would help them refresh”*
(Senior management).

This could be outlined by the SDT, which examines a range of phenomena that motivate people to act [82,83]. For instance, the SDT suggest there are three basic human psychological needs: (1) competence, where an individual has a need to achieve things and be able to take challenges, (2) autonomy, which includes being self-directed or being in control; and (3) relatedness, which includes having a mutual connection with others and a sense of belonging [69,82]. The three needs being met contribute to an individual’s motivation towards PA, health, and wellbeing as long as the needs are satisfied [70,82]. The consideration of contentment is achieved when competence, autonomy, and relatedness are supported [69]; in contrast, it can lead to a negative health impact when not supported [84]. SDT takes a multidimensional approach, highlighting why some people are motivated to change their behaviour and others are not [85]. For instance, there are different types of motivations, including intrinsic motivation, in which individuals undertake something because it is inherently exciting and enjoyable, whereas extrinsic motivation refers to things driven from the outside, such as doing something for rewards [82,83]. Therefore, if employees sense that their needs relate to competency, autonomy, and relatedness are met, it can contribute to the intrinsic motivations and result in further positive PA behaviour [82,83,86]. Thus, having autonomy can lead to intrinsic motivation and satisfaction, ultimately leading to active behaviours.


*“I don’t have that freedom to go and do some exercise when I want as we tend to be in academic services and our hours are kind of set you do kind of 9-5, so if there were some freedoms during my working hours I would exercise”*
(Professional services).

Staff recognising and suggesting ways that may help them change from sedentary to active behaviour demonstrates they are thinking about the change. Hence, they might be in the contemplation or preparation stage of the TTM and may need supportive interventions to help them move into the active phase of behaviour change. Moreover, as the SEM suggests, policies, rules, management, and overall working environment influence employees behaviour and attitudes towards their health and wellbeing [72]. Previous research has suggested that workplace policies, culture, and staff flexibility significantly affect individuals’ PA and health behaviour [11]. Advocating physical inactivity can be improved if management, policymakers, public health experts, and leisure recreation sectors embrace challenges, lead by example, and provide tailored PA and health interventions to employees in the workplace [15]. Staff are likely to be productive, positive, loyal, and lead an active and healthy lifestyle if their workplace and management are supportive [69,70].

## 5. Limitations of Study

This study found its strengths in obtaining insight into the barriers and facilitators of PA behaviours from university employees’ perspectives in according to their PA levels between job roles and genders. However, this study is not without limitations, including not exploring participants’ education levels, ethnicity, and age, which might have influenced their perceptions. Despite the limitations, the current findings provide meaningful and transferable results regarding the under-researched population’s barriers and facilitators to PA behaviour in the workplace.

## 6. Conclusions

This study aimed to understand the barriers to and potential identified facilitators of PA from university employees’ perspectives. As indicated, individuals in management positions do not take time out for lunch, and lack of management support was stated as one of the main barriers between job roles and genders preventing staff from engaging in PA. This demonstrates the unhealthy behaviour led by the senior management that also shows the unhealthy workplace culture in the participants university in this study. The unhealthy culture may negatively impact employees’ PA behaviour, retention, productivity, sickness absence, job satisfaction, and long-term health and wellbeing. To facilitate PA engagement, a comprehensive approach is needed to accommodate PA in the workplace for different job roles and genders, starting from senior management. The outcome of this study suggests that future research must focus on comprehensive interventions to promote active lifestyle workplace culture. Future research must also focus on providing educational, motivational, and informative interventions concerning the importance of engaging in PA, its effect on health, wellbeing, and the alternative ways to engage in PA whilst at work.

Having access to gym onsite, various activity classes such as walking, and sport-related competitions would be beneficial and may serve as a facilitator and contribute to employees’ PA behaviour. Additionally, management support and having flexibility in the workplace are considered key barriers, and having a protected time of 15–20 min for PA engagement may help employees engage in active workplace behaviour. This study identified generalisable and specific barriers and facilitators from university staff perception about PA engagement in the workplace. The findings contribute to the scarce literature in terms of evaluating barriers and facilitators in the workplace. The present findings can inform future practices such as PA, sedentary behaviour, health, and wellbeing-related interventions aiming to reach university employees and the broader working environment. The current findings provide meaningful and transferable results regarding university employees barriers and facilitators to PA behaviour in the workplace. These outcomes can be applied to design PA, sedentary behaviours, health, and wellbeing interventions for the under-research population in the workplace.

## Figures and Tables

**Figure 1 ijerph-19-09442-f001:**
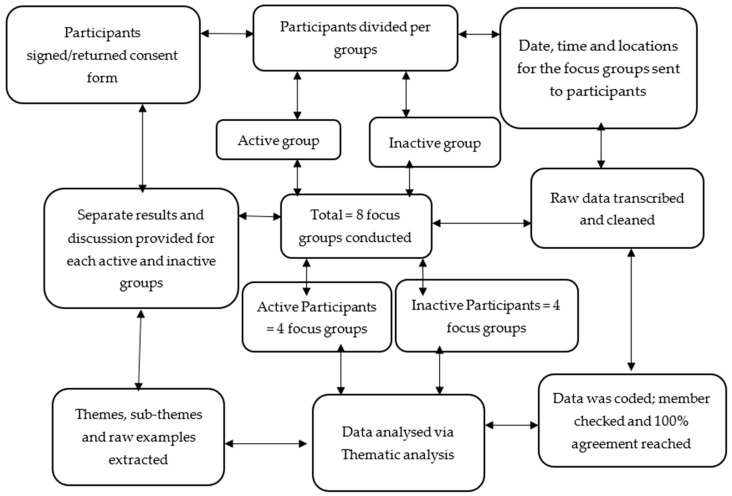
The research processes.

**Table 1 ijerph-19-09442-t001:** The breakdown of participants by gender and groups.

	Males		Females	
Total participants	Active males	inactive males	Active females	Active females
41	13	11	7	10

**Table 2 ijerph-19-09442-t002:** A summary of themes and examples of raw data extracts for barriers to and facilitators for PA from active and inactive groups.

Themes	Barriers to PA	Selected Quotes from Participants	Participants
General	Lack of Time	*“There is not enough time to do PA”*	33
	Family	*“The family life can sometimes be pressurised, one of our children is on the autism spectrum so home life can be difficult sometimes as result of that”*	17
	Motivation	*“I exercise but sometimes I feel like I need someone on my shoulder nagging me to do it. I need that motivation to get up and do it”*	21
	Weather	*“Actually, no day looks the same for me, but with the weather in the winter I don’t have the desire to do exercise”*	12
Workplace	Workplace culture	*“* *When I walk for like two miles and say to my colleagues, I walked from here to there and they be like what? Are you crazy? Why didn’t you get a car, bus, or taxi? It’s a culture thing to be rely on cars and technology”*	24
	Management	*“It is a real struggle to get staff active because most would need an approval by their managers and that’s another big task to do”*	31
	Lunch break	*“Short of lunchtime, I have 30 min lunch break and that isn’t enough”*	21
	Flexibility	*“I am not in charge of my own timetable to be honest my timetable is dictated by what other people needs from me-students and staff”*	13
	Resources	*“If you’re encouraging people to take exercise then resources need to be there for them”*	35
Facilitators to PA	Access to gym on campus	*“We need anytime access to the gym on campus”*	37
	Activity classes	*“I am quite interested in exercise, and I enjoy it I am looking forward to more diverse form of exercise for example balance and so and so”*	27
	Flexibility	*“There are number of activities that continue at lunchtime like Departmental meetings. I often get involved in Departmental meetings at lunchtime because of our timetable, as we don’t have another time and flexibility, where we can get a number of people from different groups. Having that flexibility is very important”*	19

**Table 3 ijerph-19-09442-t003:** A summary of themes and examples of raw data extracts for barriers to and facilitators for PA from active group participants.

Themes	Barriers to PA	Selected Quotes from Participants	Participants
General	Road safety	*“I started cycling but there isn’t a safe route for cycles, and I wouldn’t start again until there is safe roads because I know how bad people drive”*	7
Facilitators to PA	Enjoyment	*“I am quite interested in exercise, and I enjoy it. Yeah, I try to use weekends for exercise but something enjoyable here at work would be very helpful”* *“I run and I enjoy it and I been doing it for two and half years and I am happy to continue doing it but if I miss it, I miss it and I miss it mentally”*	9
	Positive attitudes towards PA	*“I have positive attitudes towards PA. Yeah very interested and would love to do more whilst at work”*	5

**Table 4 ijerph-19-09442-t004:** A summary of themes and examples of raw data extracts for barriers to and facilitators for PA from inactive group participants.

Themes	Barriers to PA	Selected Quotes from Participants	Participants
Commute to work	Inactive commute	*“I commute to work every day in car and then I try to find a car park space near my office, so I don’t have to walk much”*	19
General	Environments	*“There are canals to run, and I been there with my students, but I wouldn’t recommend it for a male or female. It’s pretty intermediated and there is a security issues with the canals it’s not a fantastic place to go there especially alone”*	9
	Tiredness	*“I think people are so tired after work and they may just be looking forward to tea, dinner, and TV”*	15
	Busy life	*“Busy life at home and work yeah I don’t take particular exercise, for me that’s often problematic because I got so many other things to do”*	19
	Costs	*“Well cost is a big factor when I go to swimming, although I have the associated thing in Aston Pool it does bring the cost down but still it’s not cheap and I don’t think I can subscribe to a long term because I don’t know if I can commit to it cost wise”*	13
	Likeminded people	*“I went for a walk with colleagues once and yeah there weren’t many likeminded people, and I didn’t go again”*	11
	Pressure	*“I heard a story from a member of staff saying I fall on the kitchen because I was 3 min late. If you feel that much pressure, you know it’s going to put you off from work”*	11
	Health and wellbeing not valued	*“I think there is need for a clear steer from the senior management around staff health and wellbeing. At the moment I don’t think our health is valued”*	5
Facilitators to PA	Autonomy	*“I don’t have that freedom to go and do some exercise when I want as we tend to be in academic services and our hours are kind of set you do kind of 9-5, so if there were some freedoms during my working hours I would exercise”*	16
	Protected time for exercising	*“There should be some protected time during the day if staff wanted to do some exercise or something else that would help them refresh”*	14

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
