# Peer review of "Workplace Physical Activity Barriers and Facilitators: A Qualitative Study Based on Employees Physical Activity Levels"

_ijerph, 2022, doi:10.3390/ijerph19159442_

Round 1
Reviewer 1 Report
Please see attachment.

Author Response
Reviewer 1: Comments, Suggestions, and authors responses
In this manuscript, author(s) studied the physical activity barriers and facilitators at the workplace. This is a study with great potential. However, some concerns should must address:
Methods
- Reviewer: How many people refused to participate or dropped out? Reasons?
Response: Thank you for taking time out and reviewing this paper. More information has been added to the manuscript and see an example here: From the original pool of 259 participants who signed up to be contacted, 221 employees (85%) replied expressing their interest in the present study. Despite sending four emails as reminders, 180 employees did not return their consent forms.
- Reviewer: Where was the data collected? e.g., clinic, workplace #
Response: Thank you for taking time out and reviewing this paper. This has been added: from one of the UK Higher Education Institution (university workplace)
Reviewer: What are the important characteristics of the sample? e.g., demographic data, date of enrol. Besides, please comment the inclusion and exclusion criteria.
Response: Thank you for taking time out and reviewing this paper: More information has been added to the manuscript: Despite sending four reminders, one email per week, 180 employees did not return their consent forms. Thus, they were excluded from this study. Table 1 provides the breakdown of total participants who attended a focus group interview. The inclusion criteria were that all participants had to be adults (>18 years old), currently employed by the participant university, and has participated in earlier studies.
- Reviewer: Question guide. Was it pilot tested?
Response: Thank you for taking time out and reviewing this paper: Yes. More information has been added to the manuscript section 2.3 Pilot study: Previous research suggested the importance of pilot studies in general but piloting in qualitative research is limited 40-42. The pilot study provides an opportunity to modify, review approaches and improve experience with adopted techniques 43. Therefore, prior to the actual focus groups, two pilot focus groups were conducted in this study. A total of 13 critical peers (male n = 7; females n = 6) participated in the pilot study. Participants were divided into two groups; active participants (n = 7: 4 males; 3 females) and inactive participants (n = 6: 4 males; 2 females). The pilot focus groups were conducted with staff who were not the actual study participants.
- Reviewer: What was the duration of the interviews or focus group? Discussion - Was data saturation of interviews discussed?
Response: Thank you for taking time out and reviewing this paper: More information has been added to the manuscript: Therefore, eight semi-structured focus groups with an average duration of the interview lasting 35 minutes per focus group took place in two different campuses and within four faculties of one university. Participants held various job roles and levels such as academics, senior management, professional services, and administration staff.
During the focus groups the questions were introduced one by one to receive an adequate response from participants. The lead researcher moderated the conversation and used prompts for clarity or requests for further information when required as per previous research 49. Taking this approach-built group reciprocity created an intimacy; therefore, sensitive research topics (i.e., lack of PA in the workplace) could be discussed in-depth and participants were encouraged to share and provide supporting or contrasting statements until no new themes emerged. The researcher influence was minimised by opting to remain silent during the discussions, infrequently prompting the group when discussions reached stagnation and then followed with further questions as per the previous research 50. Thus, the saturation was reached during focus groups and the data in this study was a member checked from the raw data through to complete analysis for data saturation.

Reviewer 2 Report
See attached file.

Author Response
Reviewer 2: Comments, Suggestions, and authors responses
This is an interesting work and well-written. It addresses the current concern of PA and aims to determine the barriers and facilitators of PA in the content among university employees. There are some suggestions that may improve the quality of the work.
Minor:
- Reviewer: Keywords should be more standardized, please check and edit.
Response: Thank you for taking time out and reviewing this paper: The keywords has been standardised and added to the manuscript as: Qualitative research, Physical activity, Workplace, Barriers, Facilitators, Employees health & Wellbeing
- Reviewer: Style of the reference in the main text should be changed into IJERPH style.
Response: Thank you for taking time out and reviewing this paper. This has been amended.
Major:
-Reviewer: A workflow diagram may be helpful for the readers to understand the research processes. This may be added in Section 2 Method.
Response: Thank you for taking time out and reviewing this paper: A workflow diagram of research processed has been added to the manuscript as figure 1.
- Reviewer: What is the meaning of active and inactive participants?
Response: Thank you for taking time out and reviewing this paper: Activity categorisation was based on previous studies where participants completed the International Physical Activity Long Form Questionnaire (IPAQ-LF), wore an ActiGraph for a whole week and were deemed active if they had met the WHO PA guidelines of achieving at least 150-minutes of MVPA 9. Inactive participants were those who did not meet the WH PA guideline see 32, 33.
- Reviewer: Contribution of the work, limitations of the work, and future opportunities/implications should be discussed more in conclusion.
Response: Thank you for taking time out and reviewing this paper: section 5 has been added as Limitations of study and mor information has been added into the conclusion in section 6.
Limitation of the study
This study found its strengths in obtaining insight into the barriers and facilitators of PA behaviours from university employees perspective in according to their PA levels between job roles and genders. However, this study is not without limitations, including not exploring participants education levels, ethnicity, and age, which might have influenced their perceptions. Despite the limitations, current findings provide meaningful and transferable results regarding the under researched population’s barriers and facilitators to PA behaviour in the workplace.
Conclusion
This study aimed to understand the barriers and potential identified facilitators to PA from university employees perspectives. As indicated, individuals in the management positions do not take time out for lunch, and lack of management support was stated as one of the main barriers between job roles and genders preventing staff from engaging in PA. This demonstrates the unhealthy behaviour led by the senior management that also shows the unhealthy workplace culture in the participants university in this study. The unhealthy culture may negatively impact employees PA behaviour, retention, productivity, sickness absence, job satisfaction and long-term health and wellbeing. To facilitate PA engagement a comprehensive approach is needed to accommodate PA in the workplace for different job roles and genders, starting from the senior management. The outcome of this study suggests that future research must focus on comprehensive interventions to promote active lifestyle workplace culture. Future research must also focus on providing educational, motivational, and informative interventions concerning the importance of engaging in PA, its effect on health, wellbeing, and the alternative ways to engage in PA whilst at work.
Having access to gym onsite, various activity classes such as walking, and sport-related competitions would be beneficial and may serve as a facilitator and contribute to employees PA behaviour. Also, management support and having flexibility in the workplace are considered as key barriers and having protected time of 15-20 minutes for PA engagement, may help employees engage in active workplace behaviour. This study identified generalisable and specific barriers and facilitators from university staff perception about PA engagement in the workplace. The findings contribute to the scarce literature in terms of evaluating barriers and facilitators in the workplace. The present findings can inform future practices such as PA, sedentary behaviour, health, and wellbeing-related interventions aiming to reach university employees and the broader working environment. The current findings provide meaningful and transferable results regarding university employees barriers and facilitators to PA behaviour in the workplace. These outcomes can be applied to design PA, sedentary behaviours, health, and wellbeing interventions for the under-research population in the workplace.

Round 2
Reviewer 1 Report
The authors have improved the quality of the manuscript with your corrections.
Reviewer 2 Report
Thank you very much for the reply. I appreciate your contribution. Figure 1 was not cited in the paragraph, please add it. One block "Separate results and ... " has a format issue please check.